# Enhancing the Output Performance of a Triboelectric Nanogenerator Based on Modified Polyimide and Sandwich-Structured Nanocomposite Film

**DOI:** 10.3390/nano13061056

**Published:** 2023-03-15

**Authors:** Jiaheng Zhou, Chunhao Lu, Danquan Lan, Yiyi Zhang, Yiquan Lin, Lingyu Wan, Wenchang Wei, Yuwang Liang, Dongxin Guo, Yansong Liu, Wenyao Yu

**Affiliations:** 1Guangxi Power Transmission and Distribution Network Lightning Protection Engineering Technology Research Center, Guangxi University, Nanning 530004, China; 2Guangxi Power Grid Co., Ltd., Nanning 530023, China; 3SPIC Guangxi Electric Power Co., Ltd., Nanning 530004, China; 4School of Physical Science and Technology, Guangxi University, Nanning 530004, China; 5College of Electrical Engineering, Guangxi University, Nanning 530004, China

**Keywords:** sandwich-structured nanocomposites, polyimide, triboelectric nanogenerators

## Abstract

Recently, scientists have been facing major obstacles in terms of improving the performances of dielectric materials for triboelectric nanogenerators. The triboelectric nanogenerator (TENG) is one of the first green energy technologies that can convert random mechanical kinetic energy into electricity. The surface charge density of TENGs is a critical factor speeding up their commercialization, so it is important to explore unique methods to increase the surface charge density. The key to obtaining a high-performance TENG is the preparation of dielectric materials with good mechanical properties, thermal stability and output performance. To solve the problem of the low output performance of PI-based triboelectric nanogenerators, we modified PI films by introducing nanomaterials and designed a new type of sandwich-shaped nanocomposite film. Herein, we used polyimide (PI) with ideal mechanical properties, excellent heat resistance and flexibility as the dielectric material, prepared an A-B-A sandwich structure with PI in the outer layer and modified a copper calcium titanate/polyimide (CCTO/PI) storage layer in the middle to improve the output of a TENG electrode. The doping amount of the CCTO was tailored. The results showed that at 8 wt% CCTO content, the electrical output performance was the highest, and the open-circuit voltage of CCTO/PI was 42 V. In the TENG, the open-circuit voltage, short-circuit current and transfer charge of the prepared sandwich-structured film were increased by 607%, 629% and 672% compared to the TENG with the PI thin film, respectively. This study presents a novel strategy of optimizing dielectric materials for triboelectric nano-generators and has great potential for the future development of high output-performance TENGs.

## 1. Introduction

With the rapid development of the Internet of Things (Iot), artificial intelligence (AI) and 5th generation mobile communication technology (5G), self-powered electronic products and self-powered sensors are regarded as some of the most attractive research fields [1]. Triboelectric nanogenerator (TENG) technology has the advantages of being lightweight, convenient manufacturing, being low cost, providing efficient collection of low-frequency energy and having a wide range of materials and designs. TENG not only has high performance in the field of green energy, but also has promising applications in sustainable self-powered technology [2] and data processing for artificial intelligence sensing [3]. However, the application of TENGs is mainly limited by the performances of dielectric materials. The surface charge density (σ) of the material largely determines the output performance of the TENG, which is strongly influenced by environmental conditions. Hence, finding high-output TENG dielectric materials with high strength, high wear resistance and high temperature stability is crucial.

In traditional TENG dielectric materials, the polymer and metal play a dominant role [4]. A polyimide (PI) film is commonly used as dielectric material in TENGs due to its excellent thermal stability and excellent mechanical properties [5,6,7,8,9,10]. However, traditional PI films are characterized by low electricity-generating efficiency and difficulty in processing, which limits the wide commercial application of TENGs. Researchers have used various methods to improve the output performances of TENG dielectric materials, mainly focusing on material optimization. Li et al. used inductively coupled plasma etching to modify the surface of a polyethylene terephthalate (PET) film [11,12,13], resulting in a significant increase of about 300% in terms of open-circuit voltage, short-circuit current and induced charge. Wu et al. enhanced the friction polarity by changing the functional groups on the surfaces of lignocellulose fiber materials, and introduced different functional groups to enhance the charge trapping ability [14,15,16,17]. Chen et al. introduced nanoparticles into polydimethylsiloxane (PDMS) films to optimize the surface micro-nanostructures, electron gain abilities and surface dielectric constants of the materials [18,19,20]. Zhang et al. added a layer of aluminum film between two PDMS films to improve the output performances of TENG devices. New structural designs, such as three-layer and four-layer separation structures and sandwich structures, can effectively improve the output performance [21,22,23]. Zaghloul et al. selected compatible filler materials to improve the mechanical and tribological performance of polymer composites [24,25,26,27,28,29,30]. Oh et al. used ZIF-8(porous materials) powder directly as a triboelectric layer, improving the output performances of the TENGs [31]. Nevertheless, information on the optimization and applications of PI films in TENG is still limited. Therefore, more efforts should be focused on the development of PI films and their application in TENGs by modifying the electrodes.

In this work, we used doped nanoparticles and a three-layer sandwich structure to optimize film modification. By adding a CaCu_3_Ti_4_O_12_ (CCTO) filler with strong dielectric properties to the dielectric material of a TENG, a sandwich nanocomposite film PI-(CCTO/PI)-PI was synthesized, which was composed of a CCTO-doped PI film (CCTO/PI) and PI film. In addition, the output performance of TENG was improved by optimizing the number of NPs doped. By testing various physicochemical characterization and electrical properties, it was proved that the multi-layer nanocomposite film material has excellent performance and promising applications in the field of TENGs. This study provides guidance for the design of TENG nanocomposite films.

## 2. Materials and Methods

### 2.1. Preparation of CCTO/Polyimide Film

As shown in Figure 1, 4,4′-oxydibenzylamine (ODA Aladdin, liquid state, China) and 1,2,4,5-benzenetetracarboxylic anhydride (PMDA Aladdin, liquid state, China) were mixed in a 1-methyl-2-pyrrolidinone (NMP Aladdin, liquid state, China) solution with an equal molar ratio and stirred with a magnetic mixer for 22 h under the condition of nitrogen filling to obtain a polyamide acid (PAA) solution. The main physical and chemical properties of raw materials can be seen in Table 1. The CCTO nanoparticles (NPs) (99.9%, diameter ~50 nm, Kela Material, powder state, China) were dispersed in NMP for 1 h and then stirred for 12 h to obtain the CCTO suspension in NMP. The resulting mixture was added into the PAA solution and stirred for 24 h to obtain the CCTO/PAA suspended mixture. Then, the CCTO/PAA suspension was coated on the film paver (AFA-2, Shanghai Pushen Inc., Shanghai, China) and heated in a vacuum drying oven (DZF-6050, Shanghai Rane Inc., Shanghai, China) for imidization. Subsequently, the CCTO/PI nanocomposite film was peeled off of the film-laying machine. The as-obtained film was put into the blast drying oven for drying treatment [32]. In order to determine the effects of the doping mass fraction on the properties of the materials, different doping concentration gradients were obtained by adding 2, 4, 6, 8, 10 or 12 wt% CCTO, and the CCTO/PI nanocomposites were renamed CCTO-2, CCTO-4, CCTO-6, CCTO-8, CCTO-10 and CCTO-12, respectively. Finally, based on the optimal electrical properties of the above prepared electrode (detailed discussion was provided below), we selected the material with 8% CCTO doping as the sandwich layer and prepared the sandwich electrode PI/(CCTO/PI)/PI by using the layer-by-layer casting technique, and named it 0-8-0.

Since the thickness of the nanocomposite film will affect the electrical output performance of a TENG, it is necessary to ensure that the films prepared have similar thicknesses. The thickness of the pure PI film and single-layer CCTO/PI nanocomposite film was about 25 μm, and the thickness of the three-layer nanocomposite film with a sandwich structure was about 75 μm.

### 2.2. Construction and Assembly of CS-TENG Samples

Two smooth and stable polymethyl methacrylate (PMMA) plates were used as the top and bottom substrates for the mechanical support of the entire TENG test sample. The prepared CCTO/PI and PI-(CCTO/PI)-PI composite films were pasted on the conductive tape and fixed on the bottom PMMA substrate as the electrode. Then, the copper foil was pasted on the top PMMA substrate, which was used as both the friction surface and the electrode. The CCTO/PI nanocomposite film’s thickness was ~25 μm; that of the PI-(CCTO/PI)-PI nanocomposite film was ~75 μm; the nanocomposite film’s size was 20 mm × 20 mm; the metal electrode thickness was ~80 μm; copper foil and conductive tape’s thickness was ~80 μm.

## 3. Results and Discussion

### 3.1. Chemical and Microstructural Characterization of CCTO/PI Nanocomposite Films

#### 3.1.1. Infrared-Spectroscopic Characterization of CCTO/PI Composite Films

The samples of CCTO particles and films were characterized by the Fourier transform infrared spectrometer of Bruker, Germany, with the attenuated total reflection infrared attachment. The test range was 4000–500 cm^−1^. The test sample was a square film with the size of 10 × 10 mm.

The infrared-spectroscopic characterization of nano-CCTO powder particles, pure PI film and CCTO/PI composite film samples with different doping mass fractions is displayed in Figure 2. As can be seen in Figure 2a, the absorption peaks of two different films can be observed at 725, 1080, 1230, 1380, 1720 and 1780 cm^−1^, respectively, corresponding to C=O bending vibration in the imide ring, symmetric and asymmetric stretching vibration of C-O-C, C-N stretching vibration and C=O symmetric stretching vibration and asymmetric stretching vibration. The benzene ring structure at 1495 cm^−1^ and the characteristic peak of C=C stretching vibration at 1600 cm^−1^ can be observed. Additionally, COOH and N-H characteristic absorption peaks in PAA did not appear at 2930 and 3745 cm^−1^, indicating that the thermoimide of PI and CCTO/PI single-layer composite films were basically complete. By comparing the peak strength of the characteristic peaks of the films in Figure 2a, a characteristic absorption peak in the range of 495–650 cm^−1^ was observed in CCTO/PI nanocomposite film. This may have been due to the existence of Ti-O stretching vibration peak of CCTO near 500 cm^−1^ (Figure 2b). Compared with PI thin films, the infrared absorption peak positions of CCTO/PI composite thin films showed no obvious change, indicating that the introduction of nano-CCTO particles does not change the PI structure in the composite thin films. In addition, by comparing the peak intensity of the above characteristic peaks in the composite films, it can be found that the peak intensity of the same characteristic peak in the composite films decreases as the CCTO doping content increases, indicating that nano-CCTO particles will weaken the role of these chemical bonds to some extent. Moreover, compared with a single-layer film, the absorption peak of a 0-8-0 sandwich structure does not change obviously (Figure 2c). All these proved that the introduction of nano-CCTO powder particles does not affect the unique chemical structure of polyimide, nor does it affect the imide reaction of the whole insulating polyimide film.

#### 3.1.2. X-ray Diffraction Characterization of CCTO/PI Composite Films

The X-ray diffractometer produced by PANalytical Company in Holland was used to characterize CCTO nanoparticles and thin films. The Kα target of Cu was selected as the test condition. The monochromatic X-ray wavelength (λ) was 1.540598 Å, the tube voltage was 40 kV, the tube current was 40 mA, the scanning rate was 0.013°/s, and test 2θ angles ranged from 10° to 90°. Preparation of the nano-CCTO powder test sample: Nano-CCTO was added into the XRD powder sample clip to make it smooth. Composite film test sample preparation: We cut the film sample into rectangular strips about 40 mm long and 10 mm wide and put them on XRD sample slides.

Figure 3 shows the X-ray diffraction results of CCTO nanoparticles, pure PI and 8%-CCTO/PI composite films at 2θ of 10°–80°. The pure PI film has a broad steamed-bun-shaped diffraction peak at around 2θ = 19°, which is caused by the scattering of PI macromolecular chains. The peak intensity of this diffraction peak is relatively high, indicating that PI macromolecular chains have a certain degree of order. After the introduction of nano-CCTO particles, the intensity of PI molecular diffraction peaks in the PI/CCTO composite film became weaker, indicating that the introduction of nano-CCTO particles would destroy the ordered arrangement of PI macromolecular chains to a certain extent. The possible reason is that the CCTO particles dispersed in PI may destroy the regular arrangement of PI molecules during the heating and solvent removal process of the composite material, thereby affecting the crystallinity of PI and increasing the amorphous phase of PI in the composite material. Then, the comparison between the diffraction peaks of the composite film and the CCTO powder shows that the diffraction peaks of CCTO appear near 2θ = 34°, 2θ = 49° and 2θ = 61°, and the corresponding XRD peaks of the film and the powder hardly changed. This indicates that the crystal structure of CCTO nanoparticles does not change during the preparation of composite films, and the original crystal shape remains unchanged. This indicates that the crystal structure of CCTO nanoparticles did not change during the preparation of the film, and the original crystal form remained unchanged. In addition, no impurity phase was detected, which also indicates that no other elements were mixed in during the doping process. However, the strength of CCTO was greatly weakened, which was due to CCTO being combined with PI. The presence of PI in the composite material led to a decrease in its peak strength in the composite material.

#### 3.1.3. Scanning Electron Microscopy Characterization of CCTO/PI Composite Films

The 300 KV field emission electron microscope (SEM) (SU8020, HitachiHightech, Japan) was used to detect the wear condition and surface roughness of polymer surface, and the doping particles in the film were observed. The composition of the composite insulation film was analyzed by means of X-ray spectroscopy (EDS). Before specimen observation, in order to ensure that the film has good electrical conductivity, the film sample should be treated with gold spraying before the test. The sample was cut into a square of 10 × 10 mm.

SEM was carried out to further investigate the microstructure and elemental composition of the pure PI film and the CCTO/PI insulating composite film doped with CCTO nanoparticle particles (Figure 4). Figure 4a–d shows the planar SEM images of the pure PI film and CCTO/PI film with different doping mass fractions. It can be seen in Figure 4a that there were no CCTO particles in the polymer matrix (PI). As shown, the nanoparticle CCTO powder particles in the PI matrix were distributed as islands of high concentrations, and the distribution varied with CCTO content. At the low doping content of 2 wt%, the nanoparticle CCTO powder particles were distributed very sparsely, and obvious spacing between different particles can be observed. When the doping mass reached 8 wt%, it was obviously observed that the nano-CCTO powder particles were tightly distributed and well dispersed in the film, without obvious agglomeration. When the doping content was higher than 12 wt%, the agglomeration phenomenon could be clearly observed (Figure 4d). To gain deeper insight into the morphological structure, the cross-sectional SEM images of the pure PI film, the CCTO/PI, single-layer, insulating composite film doped with CCTO nanoparticle particles and the sandwich-structured insulating composite film are displayed in Figure 5a–c. Obviously, there were basically no CCTO nanoparticles in the pure PI film (Figure 5a), but the CCTO nanoparticles were uniformly distributed in the film (Figure 5b). Additionally, the distribution of each layer of the sandwich structure can be seen clearly in Figure 5c. The EDS analysis images show the elements Ca, Ti and Cu were mainly dispersed on the surface of the material, exhibiting the successful preparation of the CCTO/PI insulating composite film (Figure 6b), which is consistent with the observations above. In Figure 6a, we can also see the distributions of Ca, Ti and Cu in the cross-section of the sandwich-structured film.

### 3.2. Performances of CCTO/PI and PI-(CCTO/PI)-PI Nanocomposite Films in CS-TENG

The programmable electrometer (Keithley6514, USA) was used to determine the energy output of the TENG. Based on the experimental platform of a linear motor, the distance was set to between 58 and 88 mm, the round-trip speed was set to 3 m/s and the acceleration was set to 10 m/s^2^. The open-circuit voltage was recorded by a 6514 electrometer. The experimental sample size was 20 × 20 mm.

The structure and working principle of CS-TENG are shown in Figure 7. Two contact friction electrodes caused an external load through copper wire. In the two-electrode mode, continuous contact separation motion occurs between the CCTO/PI composite insulating material film and copper foil. In step (i), when the CCTO/PI insulating composite film was in contact with Cu, the contact material was more likely to lose electrons and carry the same amount of positive charge. Once the two friction layers were physically in contact with each other, according to the principle of friction contact charging, a positive charge was formed on the surface of Cu, and an equal amount of negative charge was generated on the surface of the CCTO/PI film, and the outside of CCTO/PI film should have had a negative charge. When entering steps (ii) and (iii), there may have been a potential difference between the two contact materials while the surfaces of the materials in contact with each other were separated. Due to electrostatic induction, the charge on the dielectric surface increases the electrostatic field. At the same time, because the double electric layer was polarized at the electrode interface, and an equal number of excessive ionic electrons were formed at the interface to migrate from the positive electrode to the negative electrode, the surface of the Cu electrode drove the electrons attached to the bottom to flow to the PI negative electrode continuously through the external load, and this process continued until the entire electric field reached equilibrium. In step (iv), when the Cu retouched the surface of the CCTO/PI film, the entire process was reversed from step (iii), and electrons were returned from the negative CCTO/PI electrode to the Cu electrode through the external wire. The whole process was repeated in steps (i) to (iii), and the two electrodes were constantly contacted and separated. CS-TENG generated alternating current continuously. The accumulation of charges could make the dielectric surface frictional contact generated by the charge density gradually increase, the frictional charge between the contact materials after several periodic contact separation movements could reach a certain saturation state. Due to the inherently good insulation properties of PI, negative charges could be well combined in the film.

In the initial state, there was no charge transfer or potential difference between the two electrodes because there was no contact between the positive and negative friction materials. When the top Cu electrode was in contact with the PI film under external forces, electrostatic induction would have caused an equal number of heterogeneous charges on the PI film and the top Cu electrode. Following the incorporation of PI-(CCTO/PI)-PI layer, some of the S6 charges would have been captured and stored in the PI-(CCTO/PI)-PI layer, as shown in Figure 8i. When the external force was released, the electrons were transferred in the external circuit due to the broken potential equilibrium state (Figure 8ii), until the CS-TENG reached the potential equilibrium again (Figure 8iii). When the CS-TENG worked again under compressive forces, the electrons flowed in the opposite direction (Figure 8iv). Therefore, the periodic contact separation process drove induced electrons to flow back and forth between the electrodes to produce an AC output in the external circuit.

The test TENG composed of a CCTO/PI nanocomposite film and a copper sheet was fixed on both sides of the linear motor, and the output performance of the CCTO/PI nanocomposite film was measured and analyzed by using a 6514 electrometer, as shown in Figure 9a. The open-circuit voltage (Voc) of the doping mass fractions of different CCTO nano-powder particles were measured. It can be observed in the figure that as the CCTO doping mass fraction in polyimide increased, the electrical output performance of the test sample TENG generally presented a trend of first increasing and then decreasing. In addition, when the doping mass was 8 wt%, the CCTO/PI composite insulation film achieved the best electrical performance. Compared with pure PI, open-circuit voltage (Voc) increased from 20 to 42 V, which improved performance by 110%. The surface charge density and contact area were the main factors affecting the electrical performance of the friction nanogenerator. The surface charge density of the friction nanogenerator (TENG) can be improved by increasing the difference in work function between contact materials and the dielectric constant of the materials [33]. When the metal friction layer was in contact with the polymer friction layer, the surface charge density σ of the polymer can be correlated with the work function using the Poisson equation [34,35]:
(1)$$\sigma =\alpha \varepsilon_{0}\varepsilon_{r}(\frac{\Phi_{M}-\Phi_{I}}{\lambda })$$

The triboelectric effect is a continuously coupled process of charge and discharge. α is the dominant factor, which is 1.77 × 10^−13^ in the case of metal medium. ε_0_ and ε_r_ are the vacuum permittivity and relative permittivity of the insulating polymer, respectively; Φ_M_ and Φ_I_ represent the work functions of the metal and the insulating polymer, respectively; λ is the depth of charge injection. It can be concluded that the surface charge density of the friction layer increased with the increase in the relative dielectric constant of the insulating polymer. Due to the inherent capacitance characteristics of TENG, the relative dielectric constant of the insulating polymer was enhanced by doping high dielectric constant material calcium copper titanate (CCTO) to increase the capacitance of TENG and improve the output performance. When the CCTO/PI film was in contact with the Cu electrode, the electrostatic potential was generated, and the surface charge drifted from the storage layer to the surface to combine with the opposite charge, resulting in a large potential difference and increased output power. Due to interfacial polarization, the relative dielectric constant of CCTO can be improved by incorporating CCTO into PI. However, as the doping mass fraction increased, a larger doping mass fraction easily resulted in agglomeration in the material, and with the agglomeration of many CCTO nanoparticle particles, the electrical properties of the film also worsened. The leakage bridge was easily arranged along the polarization direction of the electric field, which increased the leakage current and offset the friction charge on the surface of the insulating polymer friction layer to a certain extent. In addition, the high doping mass fraction easily made CCTO nanoparticles migrate to the surface of the material, which made the surface become extremely rough and reduced the contact area between PI and Cu, resulting in the decline of the output performance. In addition, we also designed a sandwich nanocomposite film composed of a CCTO/PI nanocomposite film and a polyimide (PI) film. A charge storage layer was introduced to the middle of the TENG electrode film to improve the charge retention ability of the friction layer and further enhance the output performance of the TENG. From the working principle of the TENG, the amount of frictional charge determines the induced charge. Therefore, the charge retention ability or attenuation property of the friction-layer material is also a key factor affecting the final stable output performance of the TENG. When electrostatic potential is generated between the contact surface and the electrode, the surface charge moves from the storage layer to the surface and combines with the opposite charge, resulting in a potential difference. By introducing a charge storage layer, more charges can be accumulated and stored on the surface of the PI film, resulting in a larger potential difference and increased power output.

As shown in Figure 9b,c. The I_SC_ and V_OC_ of the interlayer nanocomposite thin film 0-8-0 were 10.25 μA and 141.5 V, respectively, which were 629% and 607% higher than those of the pure PI thin film. Additionally, the amount of transferred charge increased from 5.5 to 42.5 nC, an increase of 672% (Figure 9d). These results may be attributed to the fact that the surface charge of the film PI of the sandwich structure is trapped in the CCTO/PI intermediate layer, and the charge at the depth of the dielectric layer gradually escapes from these traps to improve the output power of the TENG [36]. By analyzing the above results, it can be concluded that the introduction of the giant dielectric constant calcium copper titanate (CCTO) can significantly provide the output performance of a TENG. Figure 9e shows the variation tendencies of peak output current and voltage of the TENG when resistors of diverse resistance values were connected to the circuit. Obviously, as opposed to the change in output current, the output voltage will decrease as the resistance decreases, until saturation occurs. According to the impedance-matching principle, when the resistance of the external load is equal to the internal resistance of the power supply (that is, the internal resistance of the TENG), the output power reaches the maximum. The variation in the output power density with resistance is shown in Figure 9f. In order to better understand the mechanism of output performance improvement, the working principle and theoretical model of TENG are analyzed. The TENGs all work on a combination of contact charging and electrostatic induction. In general, when two materials come in contact with each other, they produce opposite charges, depending on their relative positions in the triboelectric system. Opposite charges form on the electrodes attached to the back of the active material, resulting in a potential difference as the two layers approach/slide and move towards each other. This potential difference can be used as a driving source to generate an electric current that converts mechanical energy into electrical energy. Figure 10 shows the theoretical model of TENG working in vertical contact separation mode, in which the PI membrane of CCTONPs is added in the middle. Under the action of an external force, the top electrode and the PI surface form a charge of relative surface density (σ) but of opposite sign. The potential difference caused by the contact and separation of the upper and lower electrodes drives the electrons to flow through the external circuit with the release of external forces.

The charge density of the bottom electrode σ_T_ (x,t) varies with the contact distance x (t) between the top electrode and the film. The charge density of the top electrode is σ − σ_T_ (x,t), which is composed of a part of the generated friction charge and a part of the transfer charge between the electrodes. The intensity of the electric field in each region can be obtained by Gauss’s theorem.

Inside air gap:
(2)$$E_{\mathrm{air}}=\frac{\sigma -\sigma_{T}(x,t)}{\varepsilon_{0}},$$

Inside dielectric:
(3)$$E=-\frac{\sigma_{T}(x,t)}{\varepsilon_{0}\varepsilon },$$

Therefore, the voltage between the two electrodes can be calculated as follows:(4)$$V=E_{\mathrm{air}}x(t)+\mathrm{Ed}=\frac{\sigma -\sigma_{T}(x,t)}{\varepsilon_{0}}x(t)-\frac{\sigma_{T}(x,t)}{\varepsilon_{0}\varepsilon }d,$$
(5)$$\sigma_{T}(x,t)=\frac{\sigma (x)}{\frac{d}{\varepsilon }+x(t)},$$
where ε_0_ is the relative dielectric constant of the medium under vacuum condition. In addition, it can be concluded from previous studies that in the short-circuit state, the transmission charge Q_SC_, short-circuit current I_SC_ and open-circuit voltage V_OC_ are positively correlated with the surface charge density σ:
(6)$$Q_{\mathrm{sc}}=\frac{S\sigma x(t)}{\frac{d}{\varepsilon }+x(t)},$$
(7)$$I_{\mathrm{sc}}=\frac{dQ_{\mathrm{sc}}}{\mathrm{dt}}=\frac{S\sigma \frac{d}{\varepsilon }v(t)}{{\left(\frac{d}{\varepsilon }+x(t)\right)}^{2}},$$
(8)$$V_{\mathrm{oc}}=\frac{\sigma x(t)}{\varepsilon_{0}},$$

In the above formula, S is the friction surface area; x(t) represents the distance between the electrode and the contact material modified polyimide (CCTO/PI) and is a function of time; v(t) is the speed of contact separation between the upper and lower plates. It can be concluded from these equations that the output performance of the TENG depends largely on the dielectric constant of the triboelectric materials used and the charge density of the surface.

### 3.3. Analysis of the Mechanical Properties of CCTO/PI Nanocomposite Films

The samples were tested at room temperature using the AMETEKLS1 (USA) tensile tester. According to GB/T13542.2-2009, the mechanical properties of the composite films were tested at 50 mm/min. The sample standard was 100 × 15 mm, and the clip distance was 100 mm. We tested five groups of data, removed the test values with serious deviations, and then took the average test value as the test result. Tensile strength and elongation at break are expressed by Mpa and %, respectively.

As one of the main indexes of the material performance, PI films have good tensile strength. Figure 11 compared the tensile strength and elongation at break of a pure PI film, sandwich-structured nanocomposite film (0-8-0) and CCTO/PI nanocomposite film (CCTO-8). It can be seen in the figure that the pure PI film had excellent mechanical properties. Its tensile strength and elongation at break were 164.33 MPa and 25.1%, respectively. This is because the large molecule of PI contains a benzene ring and an imine ring with polar hydroxyl group, which make the molecules of a PI film have strong interactions, and the C-N structure in PI was conducive to molecular rotation. Therefore, the material has a certain softness and can be prepared for tough and elastic deformation of the film. After the introduction of nano-CCTO powder particles, the elongation at break and tensile strength of the overall CCTO/PI composite film and 0-8-0 sandwich-structured film decreased to a certain extent, and it was found that the tensile strength and mechanical properties of CCTO/PI nanocomposite films with different nano-CCTO contents were poor than those of a pure PI film, and the tensile properties of a sandwich-structured film was better than that of the single-layer CCTO/PI nanocomposite film. Compared with the pure PI film, the tensile strengths of the sandwich-structured film and the CCTO/PI nanocomposite film were 135.50 and 100.08 Mpa, respectively, which are decreases of 28.83 and 64.25 Mpa. Additionally, the elongation at break values of the sandwich-structured film and the CCTO/PI nanocomposite film were 17.25% and 12.8%, respectively, which are decreases of 7.85% and 12.3%. As we all know, when the doping content of the PI matrix reached a certain amount, the nano-CCTO powder particles would inevitably have produced agglomerations. These aggregates would have significantly affected the flexibility of PI molecules and reduced the orientation of each orientation unit in PI. The orientation of polymer molecules affected the tensile properties of the whole material to a certain extent. If external forces are applied to the material, the hydrogen bond structure and van der Waals force between non-oriented molecules will be destroyed, and the agglomeration of nano-CCTO powder particles would form multiple stress-concentration currents in the PI matrix. Under the action of external forces, the stress-concentration points would form microcracks, further damaging the membrane. Briefly, excessive doping content will not only affect the electrical properties of the nanocomposite film, but also seriously reduce its mechanical properties and fail to give play to its excellent comprehensive properties.

## 4. Conclusions

In this paper, a series of PI/CCTO single-layer composite films with different contents of nano-CCTO were prepared using PI as the matrix and nano-CCTO as the inorganic filler, and PI-(CCTO/PI)-PI sandwich films were prepared by a layer-by-layer casting method. The chemical and microscopic characterization and physical and mechanical properties of the films were tested. TENGs were prepared based on modified CCTO/PI composite films and a sandwich-structured film, and their properties were tested. The effects of different doping mass fractions of CCTO on the output electrical properties were also studied. The conclusions are as follows:

(1) After repeated experiments, it was demonstrated that the optimum doping of CCTO is 8 wt%, which doubles the steady-state open-circuit voltage in the CS-TENG, compared to pure PI films.

(2) By introducing a CCTO/PI film as the intermediate charge storage layer in PI, the output performance of the sandwich structured film is significantly improved. The results show that the maximum short-circuit current, open-circuit voltage and transmitted charge of the film prepared with a sandwich structure are 10.25 μA, 141.5 V and 42.5 nC, respectively, which are 7.3 times, 7.1 times and 7.7 times greater than those of traditional PI films.

(3) This study provides a simple method for the preparation of TENG electrode materials with good mechanical properties and high output properties.

## Figures and Tables

**Figure 1 nanomaterials-13-01056-f001:**
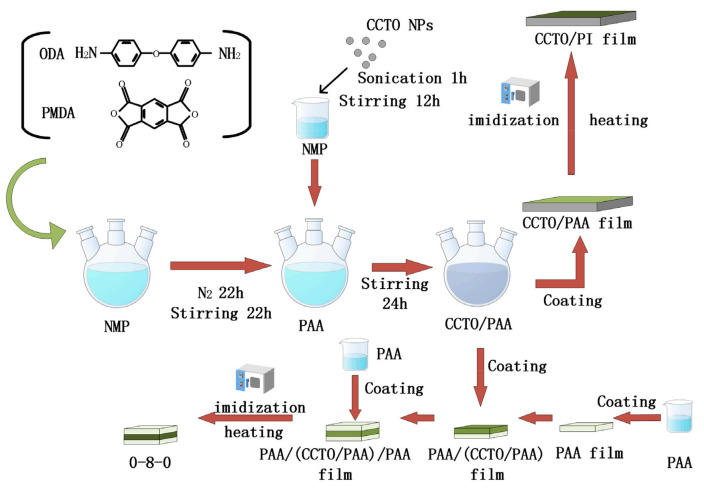
Schematic diagrams of the fabrication process for the CCTO/PI nanocomposite film.

**Figure 2 nanomaterials-13-01056-f002:**
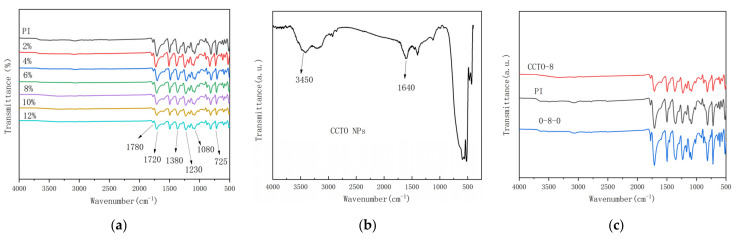
FT−IR spectra of nanocomposite films. (**a**) FT−IR spectra of pure PI film and different doping mass fractions of CCTO/PI films. (**b**) FT-IR spectra of CCTO NPs. (**c**) FT−IR spectra of CCTO/PI film, a pure PI film and a 0-8-0 sandwich structure.

**Figure 3 nanomaterials-13-01056-f003:**
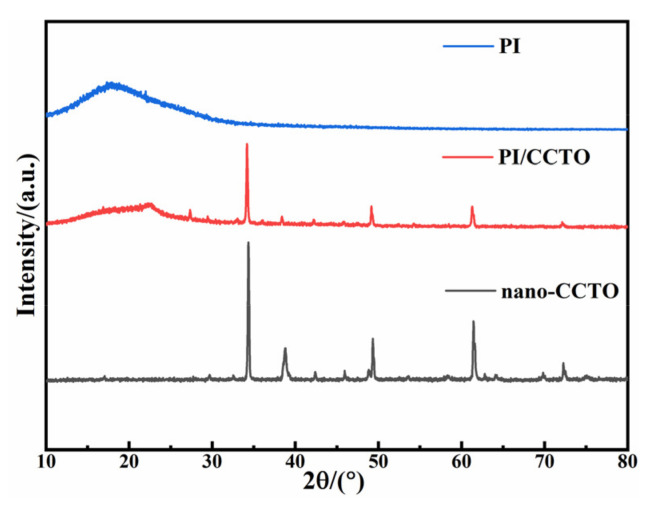
XRD patterns of pure PI, the PI/CCTO composite film and nano-CCTO.

**Figure 4 nanomaterials-13-01056-f004:**
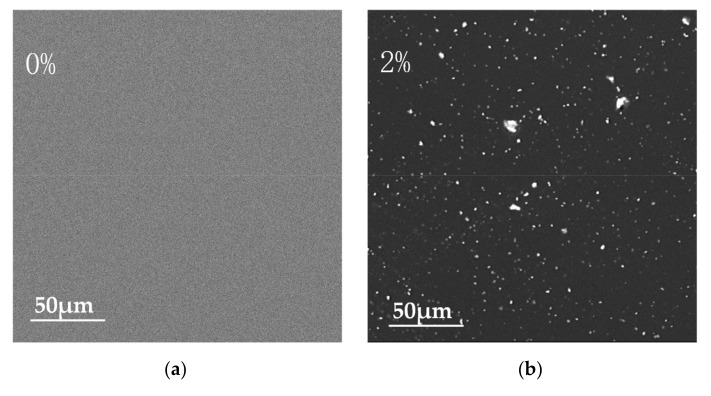
SEM images of nanocomposite film. (**a**–**d**) The planar SEM images of CCTO/PI nanocomposite films with different doping mass fractions.

**Figure 5 nanomaterials-13-01056-f005:**
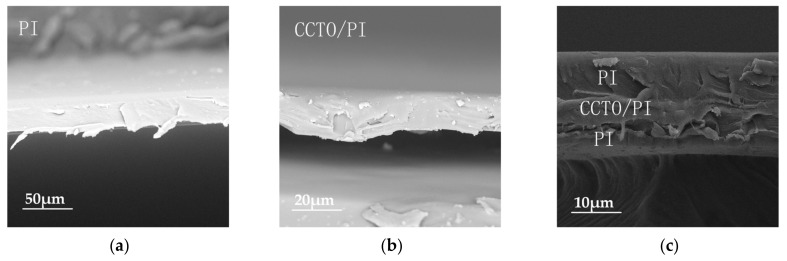
The cross-sectional SEM images of nanocomposite films. (**a**) The SEM image of a cross-section of a polyimide nanocomposite film. (**b**) The SEM image of a cross-section of a CCTO/PI nanocomposite film. (**c**) The SEM image of the cross-section of a PI-(CCTO/PI)-PI nanocomposite film.

**Figure 6 nanomaterials-13-01056-f006:**
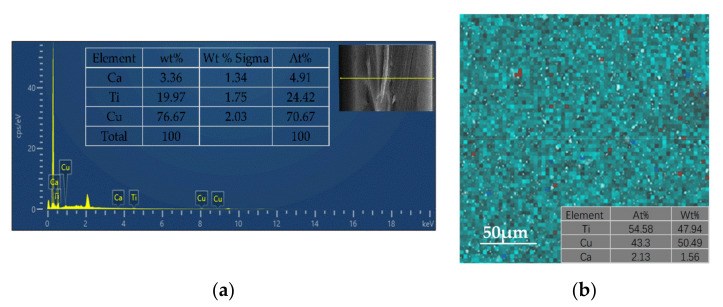
(**a**) The EDS image of a cross-section of a PI-(CCTO/PI)-PI nanocomposite film. Additionally, (**b**) the EDS images of the CCTO/PI nanocomposite film.

**Figure 7 nanomaterials-13-01056-f007:**
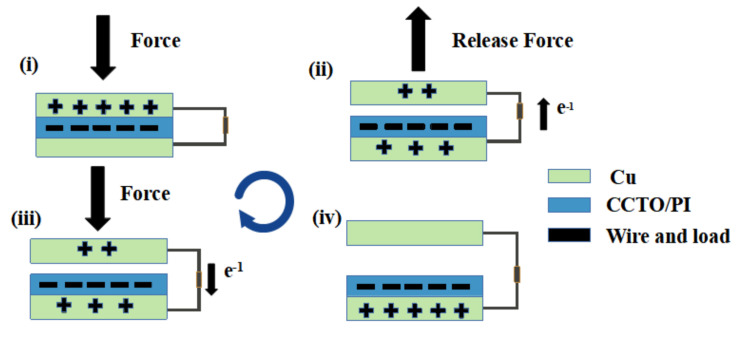
The structure and working mechanism of CS−TENG. (**i**) initial state. (**ii**) release state. (**iii**) press state. (**iv**) equilibrium state.

**Figure 8 nanomaterials-13-01056-f008:**
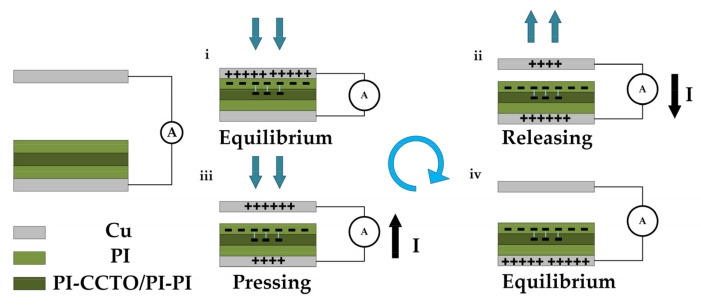
The working mechanism of CS−TENG based on a PI-(CCTO/PI)-PI nanocomposite film. (**i**) initial state. (**ii**) release state. (**iii**) press state. (**iv**) equilibrium state.

**Figure 9 nanomaterials-13-01056-f009:**
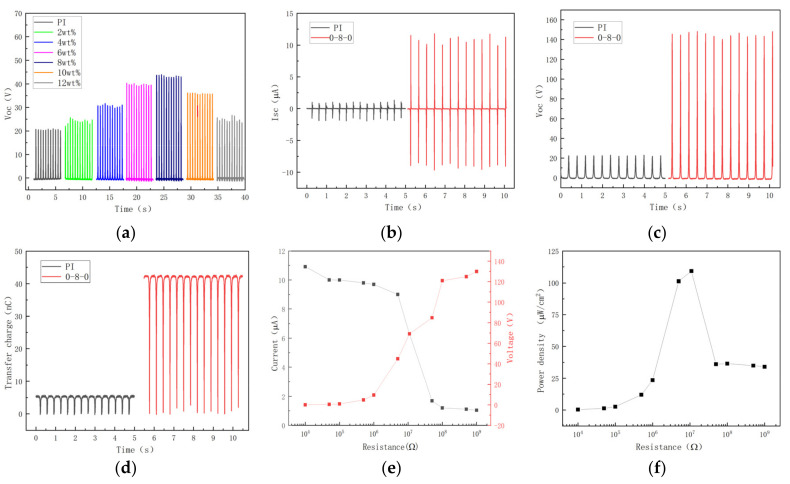
Output performances of the CS-TENG. (**a**) Variation in open-circuit voltage with doping mass fraction. (**b**) Short-circuit current (I_SC_). (**c**) Open-circuit voltage (V_OC_). (**d**) Transferred charge of PI and 0-8-0. (**e**) Current and voltage and (**f**) power density of 0-8-0.

**Figure 10 nanomaterials-13-01056-f010:**
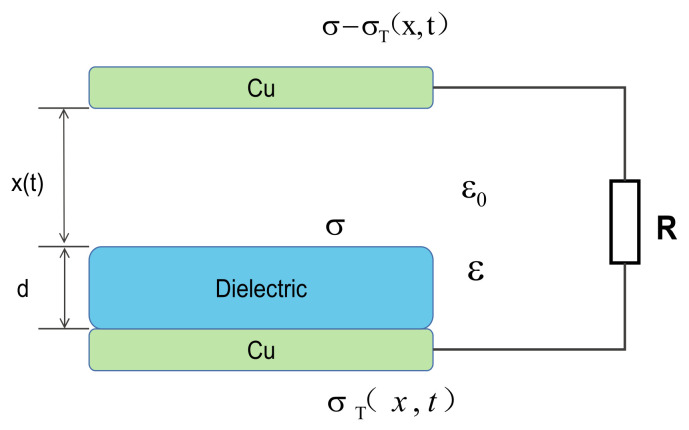
Theoretical model of CS-TENG in vertical contact-separation mode.

**Figure 11 nanomaterials-13-01056-f011:**
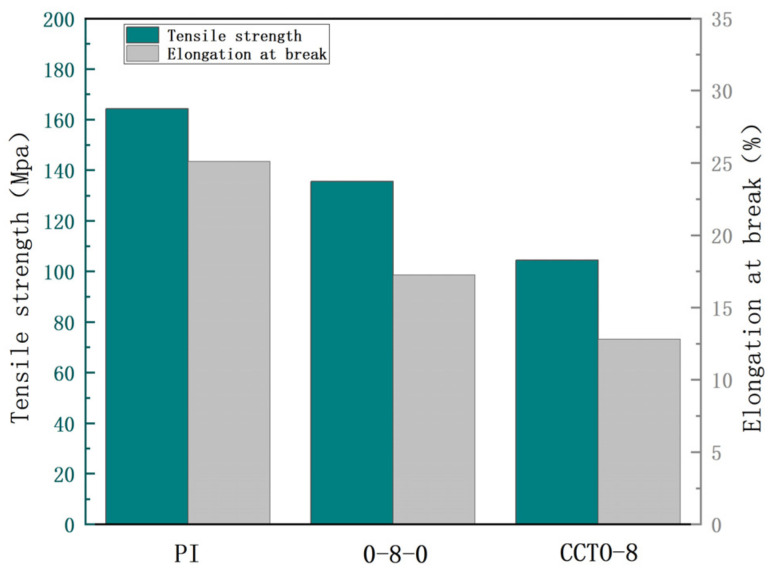
Mechanical properties analysis of nanocomposite thin films.

**Table 1 nanomaterials-13-01056-t001:** Main physical and chemical properties of the raw materials.

Material Name	Density	Acidity Coefficient
PMDA	1.68 g/cm^3^	5.45
ODA	1.11 g/cm^3^	5.49
NMP	1.028 g/mL	−0.41

## Data Availability

The study did not report any data.

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
