# Peer review of "Enhancing the Output Performance of a Triboelectric Nanogenerator Based on Modified Polyimide and Sandwich-Structured Nanocomposite Film"

_nanomaterials, 2023, doi:10.3390/nano13061056_

Round 1
Reviewer 1 Report
The paper presents an interesting approach based on the Enhancing the Output Performance of Triboelectric Nanogenerator Based on Modified Polyimide and Sandwich-Structured Nanocomposite Film. However, the innovation of the current research work should be further highlighted and emphasized. At the same time, the authors should consider the following comments to greatly improve the quality of the paper.
1. In the abstract, add a final statement that highlights the importance of this research and its possible potentials. Also, introduce the problem in the initial lines of the abstract.
2. The introduction needs to be improved by relating to the mechanics of the studied materials and their mechanical characteristics. The references to be included are: 10.1177/00219983221141154, 10.1177/0021998318790093, 10.1016/j.polymertesting.2017.09.009, 10.1016/j.compstruct.2021.114698, 10.1177/0731684417727143, 10.1002/app.46770, 10.1177/07316844211051733, 10.3390/polym15030694.
3. Kindly add a table that describes the main physical and chemical properties of the raw materials used in this study.
4. Were the preparation methods described by the authors come in accordance with a certain standard or do they follow previous procedures?
5. The schematic diagram in Figure 1 needs to be presented in a clearer version.
6. The characterization section is too small and doesn't contain any specific details. Kindly re-format it by breaking it into sub-sections. In each sub-section, the test parameters and procedures shall be explained. The sample size and geometry needs to be introduced. The testing outcomes shall be defined.
7. The mechanical performance section has only one figure that shows the tensile strength of three compositions. Where are the remaining mechanical test results?
8. The conclusion needs to be modified to summarize the research outcomes in short statements with clear observations.
Reviewer 2 Report
In this work the authors embed CCTO which has high dielectric constant into polyimide to improve performance of the triboelectric nanogenerator. They construct CCTO/PI and PI-CCTO/PI-PI sandwich structure based TENGs and compare the outputs.
Here are the reviewer comments:
1. The authors repeatedly refer to PI as electrode material which is incorrect. PI or Kapton is insulator and used an active dielectric material in TENGs. Thus the authors have mis-written "electrode" material instead of "dielectric" material in abstract, introduction and materials section. The CCTO/PI is dielectric material, not electrode. Only the Cu is electrode material. Please correct this.
2. The TENG operation depicted in Figure 5 is wrong. In (ii),(iii) the top Cu should have 2 +ve, and in (iv) the top Cu should have 0 +ve due to charge equilibrium.
3. Please draw TENG operation for sandwich structed PI-CCTO/PI-PI also.
Reviewer 3 Report
Enhancing the Output Performance of Triboelectric Nanogenerator Based on Modified Polyimide and Sandwich-Structured Nanocomposite Film written by Zhou et al aims at showing a novel strategy for optimizing electrode materials for triboelectric generators.
1. In Figure 4 the EDS spectra and the At% should be added for more detail what about the grain size CCTO ? How did the authors confirm there is no combination of CCTO particles on the polymer matrix?
2. The XRD is very important to understand whether the structure is formed or not. Please include and explain it properly.
3. In the introduction some of the important new studies based on TENG need to be discussed such as: Nano Energy, 108084,106, 2023 and Nano Energy 98, 107253, 2022
4. Figure 5 is not visible give a clear image with charge clearer? How should the CTO particle's role in improving the working mechanism be highlighted?
5. What is the output of the TENG with respect to film thickness? Give the output pattern. What is the young modulus of the films in Figure 8?
6. Why the authors did not show any characterization of CCTO materials? They have not highlighted any of its superior properties such as high dielectric constant which could possibly lead to the development of high output of TENG.
7. English grammatical errors persist throughout the introduction.
Round 2
Reviewer 1 Report
The authors have successfully done the comments. The paper can be accepted for publication.
Reviewer 3 Report
Accept as it is.